# Lactose Mother Liquor Stream Valorisation Using an Effective Electrodialytic Process

**DOI:** 10.3390/membranes13010029

**Published:** 2022-12-26

**Authors:** Arthur Merkel, Matej Vavro, Ladislav Čopák, Lukáš Dvořák, Lilia Ahrné, Christian Ruchti

**Affiliations:** 1MemBrain s.r.o. (Membrane Innovation Centre), Pod Vinicí 87, 471 27 Stráž pod Ralskem, Czech Republic; 2Department of Food Science, University of Copenhagen, Rolighedsvej 26, 1958 Frederiksberg, Denmark; 3Institute for Nanomaterials, Advanced Technologies and Innovation, Technical University of Liberec, Studentská 2, 461 17 Liberec, Czech Republic; 4Dairyfood GmbH, Göffinger Straße 6, 88499 Riedlingen, Germany

**Keywords:** desalination, electrodialysis, lactose mother liquor, lactose recovery, valorisation

## Abstract

The integrated electrodialysis (ED) process supports valorisation of a lactose-rich side stream from the dairy industry, creating an important source of milk sugar used in various branches of the industry. This work focuses on the optimization of the downstream processes before the crystallization of lactose. The process line includes a pre-treatment and desalination by ED of the industrial waste solution of the lactose mother liquor (LML). The LML was diluted to 25% total solids to overcome hydraulic issues with the ED desalination process. Two different levels of electrical conductivity reduction (70% and 90%) of the LML solutions were applied to decrease the mineral components and organic acids of the LML samples. The ED performance parameters such as ash transfer rate (*J*), the specific capacity (*C_F_*) of the ED and specific electric energy consumption (*E*) were determined and the influence of the LML solution on the monopolar ion-exchange membranes has been investigated. A higher degree of desalination is associated with higher electric energy consumption (by 50%) and lower specific capacity (by 40%). A noticeable decrease (by 12.8%) in the resistance of the anion exchange membranes was measured after the trials whereas the resistance of the cation exchange membranes remained practically unchanged. Any deposition of the alkaline earth metals on the membrane surface was not observed.

## 1. Introduction

The dairy industry produces various nutritious products with high market value. Through their production a number of nutrient-rich by-products are generated, such as whey and ultrafiltration permeates. Lactose, also referred to as milk sugar, is naturally found in the milk of mammals and belongs to a group of reducing disaccharides including glucose and galactose. This type of disaccharide is also a substrate for an uncountable number of chemical reactions, including reduction, oxidation, hydrolysis, isomerisation and biotransformation, each resulting in a different product of high interest, for example, lactitol, lactulose, or glucose-galactose syrup [1,2]. The utilization of lactose can be typically seen in food (meat, dairy, infant formula, diabetes-specific formula), confectionery, cosmetics and other industries, including synthetic fiber or glass production. Pure lactose has wide application in the pharmaceutical industry [3] as a filler or a binder to give pills suitable properties. Chemistry, mainly analytical and close natural science disciplines (biochemistry, microbiology), utilize lactose as well. Due to lactose’s plentiful usage, the global market is expected to grow substantially to 2026. Essential products driving the global lactose market are infant food and pharmaceutical drugs [4].

A number of by-products from the dairy industry, such as sweet whey, cottage cheese whey and casein whey [5,6,7] are sources of high-quality lactose. Production of high-purity lactose from whey (pharmaceutical-grade lactose with less than 0.1% ash) was described by Lifran et al. (2000) in a patented process of accelerated alcoholic crystallization. Lactose in the form of crystals is obtained by crystallization and centrifugation of the concentrated whey [6,8]. Prior to lactose production, whey can be pre-treated by various methods of protein removal techniques such as the addition of salts of calcium, ammonium, iron or polyphosphates, acids and bases, and thermal denaturation. During the production of the lactose crystals, lactose mother liquor (LML), rich in salt, needs to be desalinated to increase lactose yield and improve lactose powder quality.

Moreover, the positive effect of monovalent salts depletion on lactose yield was reported [9], and possible demineralization processes used for this step are pressure-driven processes or electrodialysis (ED) [10,11]. These separation methods have been recently favored in the industry due to their advantages in lower energy consumption compared to conventional methods of demineralization [12,13].

Pressure-driven processes and ED processes employ a solid barrier called a membrane to separate specific species from the inlet solution [14]. Demineralization of LML has been performed by nanofiltration (NF) and combined with evaporation, this results in the concentration of lactose [15]. NF also has a strong role in other branches of the food industry [16], for example, the beverage industry or saccharide recovery [17,18,19,20]. Membranes used for this size-based separation method are of composite construction with a polyamide or polysulfone selective layer [21,22]. The mechanical and chemical stabilities are excellent (temperature 50–60 °C, pH 1–13, pressure up to 0.5 MPa) [7].

As a competitive demineralization process, ED differs from NF in the transfer mechanism and driving force. Unlike NF, conventional ED does not require a high hydrostatic pressure of inlet feed solution to induce mass transfer. However, the hybrid technology of electro-filtration, which combines electrodialysis and a pressure-driven process in one separation step, was developed and applied to treat β-lactoglobulin [23].

The presence of an electric field causes ion migration in ED. Charged ions released from dissolved salts are easily removed from the solution and fed into an electrodialyzer. Uncharged species like sugars or alcohols are held in the bulk of the feed with a higher purity compared to the initial state of the feed treated. The solid barrier which facilitates ion transfer is an ion-exchange membrane. The ion-exchange membranes can be classified into various groups depending on the method of manufacture (heterogeneous, homogeneous, interpolymer) or their selective transport characteristics (monopolar and bipolar) [7,24,25,26]. Regardless of the membrane’s group, each membrane contains charged species that facilitate the transfer of the counterion.

ED has been chosen to purify many kinds of feed, including different types of whey, wine, sugars or glycols [27,28,29,30,31,32,33].

In the present study, a higher degree of desalination has been tested to estimate its influence on the process parameters compared to previous work [8], in which the ~70% removal of ash has been studied.

## 2. Materials and Methods

### 2.1. Materials and Chemicals

The lactose mother liquor (LML) for demineralization experiments was taken directly from the lactose production line in Dairyfood GmbH (Riedlingen, Germany). Following LML withdrawal, the LML was diluted with RO water in a ratio of 1:2.25 due to the high viscosity of the original LML. The obtained suspension was diluted to approximately 25% total solids (TS).

The deionized water (DW) was produced in Dairyfood GmbH (Riedlingen, Germany) by reverse osmosis (RO). The chemicals used in the experiments were of analytical grade and purchased from Merck s.r.o. (Prague, Czech Republic).

### 2.2. Membranes for Electrodialysis

Commercial food-grade anion and cation exchange membranes (AMH and CMH Ralex^®^) were used in the electrodialysis process. These are heterogeneous membranes based on polyethylene (PE) as polymer matrix and sulfonic acid groups (R–SO_3_^−^) as cation exchange, and quaternary ammonium groups (R–(CH_3_)_3_N^+^) as anion-exchange groups. Furthermore, both membranes were reinforced with polyester (PES) fabrics. The membranes were produced by MEGA a.s. (Stráž pod Ralskem, Czech Republic).

### 2.3. Electrodialysis

The ED experiments were conducted in Dairyfood GmbH (Riedlingen, Germany) using an ED unit (MemBrain s.r.o., Stráž pod Ralskem, Czech Republic). The ED unit was equipped with an ED stack consisting of 10 pairs of CMH and AMH in CMH–AMH–CMH configuration. Polyethylene spacers with a mesh and thickness of 0.8 mm were used to hydraulically separate, dilute and concentrate the chamber. Polyethylene spacers with a mesh and thickness of 1.0 mm secured hydraulic separation of the electrolyte solution on endplates of the ED stack, inserted between CMH and electrode. Distributors with a greater thickness were placed in separate electrolyte chambers due to more efficient gas extraction created close to the electrode surface. The diluate cylinder was filled with 1.0 kg of diluted LML, while concentrate and electrolyte cylinders contained 0.5 kg of deionized water and 0.25 kg of 1% (wt./wt.%) sodium nitrate solution, respectively. Throughout the experiments, the temperature of the diluate and concentrate was maintained at 15 °C by immersing stainless steel helices into process solutions, which were connected to the cryostat unit Julabo CF41 (Seelbach, Germany). Circulation flow rates of diluate, concentrate, and electrolyte solutions were set to 55 L·h^−1^, 55 L·h^−1^ and 50 L·h^−1^, which corresponds to linear velocity 4.8 cm·s^−1^ for diluate and concentrate, and 17.4 cm·s^−1^ for the electrolyte solution. A proportional-integral controller automatically regulated the flow rates while continuously measured by a magnetic-inductive flow meter IFM SM4000 (Essen, Germany). Lactose production with integrated ED technology of LML recovery is presented in Figure 1.

The applied potential difference at the membrane stack was 10 V, corresponding to a potential of 1 V per membrane pair. The pH value and conductivity of the solutions were recorded every 10 min using an automatic recording system comprised of a Schneider Electric (Schneider Electric CZ s.r.o., Prague, Czech Republic) and Endress+Hauser Liquiline (Endress+Hauser Czech s.r.o., Prague, Czech Republic), CPS71D–7TP21 glass pH sensor and CLS82D conductivity stainless steel sensor (Endress+Hauser Liquiline), respectively. Values of the voltage and resulting current were recorded every 10 min throughout the experiments. The physicochemical properties of individual samples were determined just before starting the demineralization sequence and immediately after the termination of the demineralization sequence and discharge of the individual tanks. The conductivity, pH, and temperature of collected samples were measured using WTW Multi 3620 IDS equipped with conductivity probe Tetra Con^®^925 and pH glass electrode Sentix^®^ 940 (WTW, Weilheim in Oberbayern, Germany). The cleaning-in-place (CIP) procedure was performed after all ED experiments. The CIP procedure involved cleaning the ED membrane stack with 3% (w/w%) HNO_3_ and NaOH for 20 min with water flushing for 20 min after both the acid and alkaline solutions. Conditions of the ED process setup are shown in Table 1.

### 2.4. Methods

Concentrations of inorganic ions, TS, ash content (i.e., total mineral content), and ash relative to the dry base (ODB%) were measured according to procedures described by [25]. The concentrations of cations were analyzed using optical emission spectroscopy with an inductively coupled plasma (ICP–OES) from Thermo Fisher Scientific GmbH (Munich, Germany).

The concentrations of inorganic anions were determined with ion chromatography using Dionex™ ICS–5000+ DC from Thermo Fisher Scientific GmbH (Munich, Germany), equipped with a conductometric detector.

Protein content in samples was calculated according to the molecular nitrogen content with the specific nitrogen-to-protein conversion factor of 6.38. For determination of nitrogen content, 2 g of sample was kept at 900 °C for 7 min. Products of combustion flow through a sorption column, where CO_2_ and H_2_O are removed. After CO_2_ and H_2_O removal, gas flows through a reduction column, where NO_X_ substances are reduced to a molecular nitrogen N_2_. The molecular nitrogen content is subsequently evaluated with gas chromatography using a thermal conductivity detector (TCD). The values of the nitrogen content were obtained using rapid MAX N exceed (Elementar, Langenselbold, Germany).

Scanning electron microscope (SEM) images were obtained using a scanning electron microscope Quanta FEG 450 (FEI, Hillsboro, OR, USA) at 10 kV accelerating voltage, magnification at 300× and 80 Pa residual pressure. The thickness of each sample was taken as the average value of five points measured before the experiments by a Mitutoyo micrometer. A detailed description of ion exchange capacity measuring and calculation can be found elsewhere [34].

The citric acid (CA) and lactic acid (LA) concentrations were determined using the capillary isotachophoresis technique with a one-purpose analyzer [35].

The lactose (LAC) content was determined by a polarimeter from A. Krüss Optronic GmbH, model “P1000 LED” (Hamburg, Germany). Besides LAC, the optically active compounds were precipitated by adding potassium ferrocyanide and zinc sulphate solutions, followed by precipitate filtration [36]. The filtrate was subsequently subjected to polarimetric analysis.

The density of samples was measured by a digital density meter Densito 30Px (Mettler Toledo, Columbus, OH, USA).

The apparent permselectivity and resistivity of the membranes were measured according to methods described elsewhere [37]. The content of inorganic cation in membranes was measured by X-ray energy-dispersive (EDX) spectrometer Team Software Suite, coupled with the Octane Elect and Octane Elite at 10 kV on a scanning electron microscope (SEM) Quanta FEG 450, FEI, USA.

### 2.5. Calculations

The removal of ions *R* (%) in diluate during ED was calculated according to Equation (1).
(1)$$R=\left(1-\frac{V_{p}c_{p,i}}{V_{f}c_{f,i}}\right)\cdot 100\%$$
where, $V_{p}$ (L) and $V_{f}$ (L) are the volumes of the diluate after and before ED, $c_{p,i}$ (mg·L^−1^) and $c_{f,i}$ (mg·L^−1^) are concentrations of the respective compound after and before ED.

Considering a hypothetical compound A_p_B_q_, the dissociation equilibria are described by Equation (2).
(2)$$K_{\mathrm{sp}}\left(A_{p}B_{q}\right)=\frac{a{\left(A^{q+}\right)}^{p}\cdot a{\left(B^{p-}\right)}^{q}}{\left(A_{p}B_{q}\right)}$$
where, $K_{\mathrm{sp}}\left(A_{p}B_{q}\right)$ is the solubility product of a hypothetical compound A_p_B_q_; *a*(A^q+^), *a*(B^p+^) and *a*(A_p_B_q_) are the ionic activities of A^q+^, B^p+^ species and a hypothetical compound A_p_B_q_, respectively and p^+^ and q^−^ are electric charges of respective ions.

Conductivity reduction $\kappa_{\mathrm{CUT}}$ (%) of diluate for each experiment was calculated by Equation (3).
(3)$$\kappa_{\mathrm{CUT}}=100\cdot \left(1-\frac{\kappa_{D}}{\kappa_{F}}\right)$$
where, $\kappa_{\mathrm{CUT}}$ (%) is the conductivity reduction of diluate, $\kappa_{D}$ (mS·cm^−1^) is the final conductivity of diluate, and $\kappa_{F}$ (mS·cm^−1^) is the initial conductivity of diluate.

Ash transfer rate for each demineralization degree was calculated by Equation (4).
(4)$$J=\frac{m_{F}w_{\mathrm{Ash},F}-m_{D}w_{\mathrm{Ash},D}}{N\cdot w\cdot l\cdot ∆t\;}$$
where, *J* (g_Ash_·m^−2^·h^−1^) is the ash transfer rate, $m_{F}$ (kg) and $m_{D}$ (kg) are mass of feed and diluate, respectively, $w_{\mathrm{Ash},F}$ (wt./wt.%) and $w_{\mathrm{Ash},D}$ (wt./wt.%) are mass fractions of ash in the feed and diluate respectively, *N* (–) is the number of installed membrane pairs, *w* (m) is the active width of the ion-exchange membrane, *l* (m) is the active length of the ion-exchange membrane, and ∆*t* (h) is the duration of the experiment.

The specific capacity of the ED unit *C*_F_ (kg_F_·m^−2^·h^−1^) is calculated by Equation (5).
(5)$$C_{F}=\frac{m_{F}}{N\cdot A\cdot ∆t}$$
where, $C_{F}$ (kg_F_·m^−2^·h^−1^) is the specific capacity of the ED unit, $m_{F}$ (kg) is the mass of the feed, *N* (–) is the number of installed membrane pairs, *w* (m) is the active width of the ion-exchange membrane, *l* (m) is the active length of the ion-exchange membrane, and ∆*t* (h) is the duration of the experiment.

The specific consumption of electric energy for ion transport *E* (Wh·kg_F_^−1^) is calculated by Equation (6).
(6)$$E=\frac{\left(U_{\mathrm{Avg}}\;\int_{0}^{t}Idt\right)}{m_{F}}\;$$
where, *E* (Wh·kg_F_^−1^) is the specific consumption of electric energy for ions transport, U_Avg_ (V) is the average potential difference applied to the membrane stack, $\int_{0}^{t}Idt$ (A·h) is the amount of transferred electric charge, and $m_{F}$ (kg) is the mass of the feed.

The specific water consumption for concentrate dilution $m_{W}$ (kg_W_·kg_F_^−1^) is calculated by Equation (7).
(7)$$m_{W}=\frac{m_{W,i}-m_{W,f}}{m_{F}}\;$$
where, $m_{W}$ (kg_W_·kg_F_^−1^) is the specific water consumption for concentrate dilution, $m_{W,i}$ (kg) is the initial mass of water in the reservoir, $m_{W,f}$ (kg) is the final mass of water in the reservoir, and $m_{F}$ (kg) is the mass of the feed.

Specific consumption of concentrated acid for concentrate pH adjustment $m_{{\mathrm{HNO}}_{3},65\%}$ (g_Acid_·kg_F_^−1^) is calculated by Equation (8).
(8)$$m_{{\mathrm{HNO}}_{3},65\%}=\frac{m_{\mathrm{Acid},i}-m_{\mathrm{Acid},f}}{m_{F}}\frac{w_{{\mathrm{HNO}}_{3},\mathrm{dil}.}}{w_{{\mathrm{HNO}}_{3},\mathrm{conc}.}}\;\;$$
where, $m_{{\mathrm{HNO}}_{3},65\%}$ (g_Acid_·kg_F_^−1^) is the specific 65% (wt./wt.%) acid consumption for concentrate pH adjustment, $m_{\mathrm{Acid},i}$ (g) is the initial mass of 3% (wt./wt.%) acid in the reservoir, $m_{\mathrm{Acid},f}$ (g) is the final mass of 3% (wt./wt.%) acid in the reservoir, $w_{{\mathrm{HNO}}_{3},\mathrm{dil}.}$ (wt./wt.%) is the mass fraction of diluted acid in the reservoir, $w_{{\mathrm{HNO}}_{3},\mathrm{conc}.}$ (wt./wt.%) is the mass fraction of concentrated acid, and $m_{F}$ (kg) is the mass of the feed.

Using the equilibrium dissociation constants for an *n*-protic acid and balancing individual products of the dissociation, it is possible to derive the following equations for the molar fraction of individual species in a solution, demonstrated by Harris et al. [38].
(9)$$\alpha_{H_{n}A}=\frac{{\left[H^{+}\right]}^{n}}{D}$$
(10)$$\alpha_{H_{n-1}A}=\frac{{\left[H^{+}\right]}^{n-1}K_{1}}{D}$$
(11)$$\alpha_{H_{n-j}A}=\frac{{\left[H^{+}\right]}^{n-j}\prod_{i=1}^{j}K_{i}}{D}$$
(12)$$D={\left[H^{+}\right]}^{n}+{\left[H^{+}\right]}^{n-1}K_{1}+{\left[H^{+}\right]}^{n-2}K_{1}K_{2}+\cdots +{\left[H^{+}\right]}^{n-j}\prod_{i=1}^{j}K_{i}$$
where, $\alpha_{H_{n}A}$, $\alpha_{H_{n-1}A}$ and $\alpha_{H_{n-j}A}$ are molar fractions of H_n_A, H_n−1_A and H_n−*j*_A species, [H^+^] is the activity of protons, n and *j* denote the number of donating protons and number of equilibrium dissociation constants, respectively. $K_{1}$, $K_{2}$, …, $K_{j}$ represent the equilibrium dissociation constants of first, second, and up to *j*-th dissociation equilibria. D is a parameter dependent on the number of H^+^ ions of the considered acid. In Equations (9)–(12), it is possible to substitute the activity of H^+^ ions with a pH value to get the direct relation between the molar fractions of the respective compound in a solution and the pH of the solution. The thermodynamical definition of the equilibrium dissociation constant implies that it is a function of ionic strength and temperature. Thus, the equilibrium dissociation constant, which corresponds to the ionic strength and temperature of the processed solution, should be used to model molar species distribution, as demonstrated by Mizera et al. [39] and Reijenga et al. [40]. For the molar species distribution modelling of CA and LA, we have used equilibrium constant values demonstrated by Martell et al. [41], which are valid for infinitely dilute aqueous solutions (*I* = 0 mol·L^−1^) up to ionic strength *I* = 2.0 mol·L^−1^ and temperature of 25 °C. During the demineralization, the ionic strength of the processed solution continually decreases due to ion removal. Thus, the ionic strength of the feed and LML R90 was calculated using Equation (13) to evaluate the ionic strength shift. The average ionic strength value was used to select the equilibrium constant *K_i_* and calculate the molar species distribution.
(13)$$I=\frac{1}{2}\overset{n}{\underset{i=1}{\sum }}c_{i}z_{i}^{2}\;\;$$
where, *I* [mol·L^−1^] is the ionic strength of the solution, $c_{i}$ [mol·L^−1^] is the molar concentration of a specific ion, and $z_{i}$ [–] is the electric charge of the respective ion. Since the tabulated values are valid for the temperature of 25 °C, it was necessary to evaluate the equilibrium constant value at the temperature of 15 °C, at which the demineralization of LML was conducted. Using the Van’t Hoff equation it is possible to calculate an equilibrium constant $K_{2}$ at temperature $T_{2}$ if the equilibrium constant $K_{1}$ at temperature $T_{1}$ is known (Equation (14)).
(14)$$\log K_{2}=\log K_{1}+\frac{∆H}{2.303\;R}\left(\frac{1}{T_{1}}-\frac{1}{T_{2}}\right)\;\;$$
where, $K_{2}$ [–] is the equilibrium constant at temperature $T_{2}$ [K], $K_{1}$ [–] is the equilibrium constant at temperature $T_{1}$ [K], ∆*H* [kcal·mol^−1^] is the enthalpy change of dissociation and R [kcal·mol^−1^·K^−1^] is the universal gas constant.

Regarding Equation (14), a simplification was adopted, and the ∆H is considered constant for the temperature range of 15–25 °C.

### 2.6. Statistical Analysis

Data for feed and product quality are averages (*N* = 4) with a corresponding confidence interval, calculated by the Student *t*-test with a confidence level α = 0.05.

## 3. Results

### 3.1. Electrodialysis

Figure 2a,b shows the changes in the conductivity and current during the desalination of the samples. As observed, differences in the rates of demineralization are mainly attributed to membrane fouling, which decreased the demineralization rate concerning a product with specific ions’ removal. Besides that, minor differences in initial temperatures of the feed stream (diluate) due to practical experimental limitations could also have some minor influence on the rates of demineralization. During the demineralization of LML, specific phenomena were observed, characteristic of the ED process, such as non-zero current value although no voltage was applied to the membrane stack [8]. This phenomenon is explained by the chemical potential difference between the diluate and concentrate chambers. Thus, ions are spontaneously transported against the concentration gradient through the ion-exchange membrane by a diffusion mechanism, and a non-zero current value is observed. The value of the electric current increases in the first 20–30 min of demineralization, which relates to the concentrate conductivity increase due to ions transferred from the diluate chamber. After the electric current reaches its maximum value, a current decrease simultaneously with a diluate conductivity decrease is observed. This observation is explained by the further depletion of ions from the diluate to the concentrate. Further depletion of ions from the diluate increases diluate electric resistivity, while the concentrate electric resistivity decreases due to receiving ions.

However, the conductivity and pH of the concentrate were maintained below 15.0 mS∙cm^−1^ and pH 5.5 by adding deionized water and 3% (wt./wt.%) HNO_3_, respectively. Maintaining the conductivity and pH of the concentrate below the stated values avoids reaching the solubility limit, defined by the solubility product $K_{\mathrm{sp}}$ (Equation (2)), of low soluble salts in the general form of M_x_(H_y_PO_4_)_z_ and MCO_3_ (M=Ca^2+^, Mg^2+^). Samples of the demineralized product were taken for analysis when the diluate conductivity reduction of 70% and 90% was reached in individual experiments.

The ED performance coefficient parameters of the two products are presented in Table 2.

### 3.2. Minerals and Organics Removal Efficiency

Individual ions in the electrodialysis process are transported from the diluate, through the ion-exchange membranes, to the concentrate compartment by diffusion, osmosis, electro-migration and electro-osmosis [42]. From a practical point of view, ions’ removal using electrodialysis is beneficial for evaporator performance due to decreasing the scaling potential of low soluble inorganic salts on the heat exchange surface of an evaporator, such as calcium phosphate Ca_3_(PO_4_)_2_ ($K_{\mathrm{sp}}$ = 10^−25.5^–10^−24.8^). Minimizing the scaling potential results in an increase in the evaporator performance and reduces CIP frequency, as well as operational expenses related to purchasing CIP chemicals and wastewater management. Besides the ions of inorganic salts, dissociated organic acids and their corresponding salts are also reduced in the electrodialysis process. The removal of lactic acid and calcium lactate is essential in terms of downstream processing of the demineralized mother liquor in a further crystallization and spray-drying tower. It was investigated by Chandrapala et al. [43] that the presence of lactic acid and calcium negatively affects the effectivity of lactose crystallization and causes stickiness of the spray-dried products. Thus, implementing the electrodialysis process improves the effectivity of further mother liquor processing and the quality of the final product.

The removal efficiency of monovalent ions such as K^+^, Na^+^ and Cl^−^ is higher than multivalent ions such as Ca^2+^, Mg^2+^, SO_4_^2−^ and PO_4_^3−^. Regarding the removal efficiency, the following order was observed for cation ions: K^+^ > Na^+^ > Mg^2+^ > Ca^2+^ and for anion ions: Cl^−^ > SO_4_^2−^ > PO_4_^3−^ > LA^−^ > CA^3−^, which corresponds to the observation of the previous work [8]. The difference of removal efficiency could be the LML solutions, which have significantly different mineral and organic profiles, mostly in their lactates and citrates content. Moreover, the quality of the feed material and the technological strategy of lactose production may also impact the removal efficiency. These orders indicate that the K^+^ ions were depleted the most, while the Ca^2+^ ions were depleted the least. Concerning anions, the Cl^−^ ions were depleted the most, while the CA^3−^ ions were depleted the least. This phenomenon is explained by a smaller hydrodynamic radius and higher diffusion coefficient of monovalent ions compared to the multivalent ions [44,45]. Overall, the total content of inorganic ions (ash) was decreased by 65.0% ± 1.5% (wt./wt.%) and by 82.3% ± 0.3% (wt./wt.%) in the case of desalted LML R70 and LML R90, respectively. Furthermore, for organic acids, the removal efficiency depends on the pH of the processed solution, which affects the ionization of organic acids to related species in a solution. Using dissociation constants of specific acid, the distribution of ionized species in solution is calculated at defined pH, ionic strength and temperature, Figure 3a,b [40]. The most efficient removal of organic acids is reached when the pH of the processed solution is in a range where the ionized form of organic acids prevails in the solution [11].

Figure 3a shows that the citric acid is wholly dissociated with the corresponding species in the operating pH range of diluate (pH 5.12–4.76). In contrast, lactic acid (Figure 3b) is presented in non-dissociated form, ranging from 3% at pH 5.12 to 8% at pH 4.76. However, the removal efficiency of citrates was not as significant as that of lactates, which is explained by the higher hydrodynamic radius and lower diffusion coefficient of the citrate anions compared to the lactate anion. In comparison, the absolute content of lactates was reduced by 42.0% ± 3.0% and 78.4% ± 2.2% for the LML R70 and LML R90, respectively, while the absolute citrate content was reduced by 15.7% ± 6.8% and by 34.0% ± 1.8% for the LML R70 and the LML R90, respectively. The average composition of diluted LML (feed), demineralized LML R70 and LML R90 are presented in Table 3.

The removal of the main components of final products LML R70 and LML R90 is shown in Figure 4.

### 3.3. Membrane Properties and Energy-Dispersive X-ray Spectroscopy (EDX)

After the ED process, a decrease in resistance and permselectivity of AMHs is observed (Table 4), which can be attributed to the expansion of pores and ionic channels under the water flow pressure. For CMHs, on the contrary, permselectivity increases while the resistance remains within the margin of error. The increase in permselectivity may be due to contamination of the membrane surface or pores by organic molecules, such as proteins or inorganic low soluble salts, such as calcium or magnesium phosphates of the composition M_x_(H_y_PO_4_)_z_.

However, according to SEM micrographs, the membrane surface on the diluate side is smooth with no prominent salt deposits (Figure 5), and EDX spectroscopy of the CMHs surface shows no evidence of calcium or magnesium phosphates (calcium and magnesium atomic contents are 0.4 ± 2.0% and 0.1 ± 9.4%). This may be due to the fact that in the CIP process, the salt deposits are dissolved. Thus, the most likely increase in the permselectivity of the CMHs after electrodialysis is due to the fouling of the membrane surface by proteins, which under conditions of pH~5.1 are presented as positively charged molecules since their isoelectric points are below this pH value.

## 4. Conclusions

This study shows one possible utilization of lactose mother liquor (LML), which is a waste by-product in the process of lactose crystallization, using an effective electrodialysis process. Demineralized LML can be recycled in the process of lactose manufacture or spray-dried and used for animal feed due to its high lactose content and proteins. Electrodialysis decreases the ash content in the dry matter of the LML solution, which is similar to sweet whey (approx. 7–8 ash, ODB%). The partially demineralized mother liquor can be mixed with the sweet whey, resulting in an increase of total solids and lactose as well as the improvement of the lactose yield during the crystallization process.

The chemical composition of products with different degrees of demineralization was investigated, as well as the performance parameters of the electrodialysis process, such as ash transfer rate, specific capacity, electric energy, water and chemical consumption. The absolute ash content in the LML was reduced by 65.0 ± 1.5% and 82.3 ± 0.9% for the LML R70 and LML R90, respectively. Moreover, the efficiency of removal of organic acids was investigated. The absolute content of lactic acid was reduced by 42.0 ± 2.9% and 78.4 ± 2.2% for LML R70 and LML R90, respectively, which is of great practical importance in increasing lactose crystallization yield and improving the properties of the spray-dried products. It was demonstrated that the removal efficiency of organic acids depends on the pH, temperature, composition and concentration of individual components in the processed solution.

Furthermore, the energy demands for the salt transfer to produce LML R70 and LML R90 are 20.60 ± 1.17 Wh·kg_F_^−1^ and 30.03 ± 3.41 Wh·kg_F_^−1^ (DC), respectively. The consumption of demineralized water for the concentrate conductivity make-up, to prevent precipitation of low soluble salts, was 2.62 ± 0.09 kg_W_·kg_F_^−1^ and 3.48 ± 0.30 kg_W_·kg_F_^−1^ for LML R70 and LML R90, respectively.

The physicochemical properties of the IEMs were investigated and compared before and after the demineralization trials. It was found that the specific electric resistivity and permselectivity of the AMHs were decreased, contributing to pore and ionic channel expansion due to water flow pressure. On the contrary, the specific electric resistivity of the CMHs remains within the margin of error while the permselectivity increases, which is explained by the membrane surface fouling by positively charged proteins. The EDX spectroscopy showed no evidence of calcium and magnesium phosphates on the surface of the CMHs.

## 5. Patents

Lifran, E.V.; Sleigh, R.W.; Johnson, R.L.; Steele, R.J.; Hourigan, J.A.; Dalziel, S.M. Method for purification of lactose. WO2002050089A1. Published 27 June 2002.

## Figures and Tables

**Figure 1 membranes-13-00029-f001:**
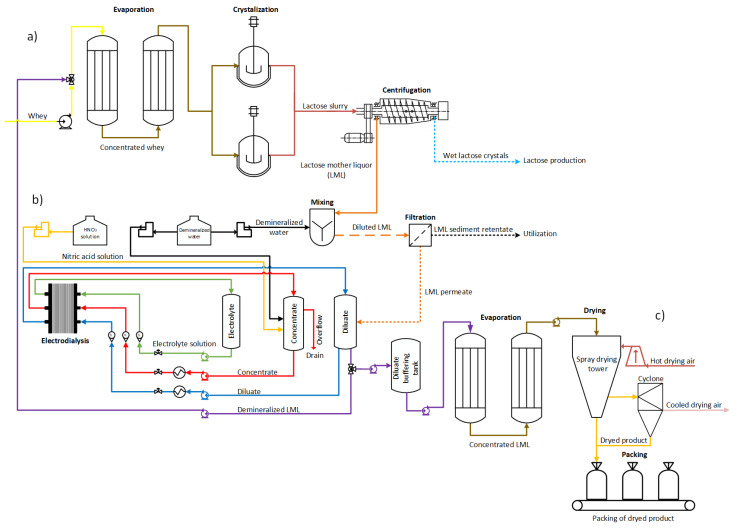
Scheme of the experiments. Lactose production (**a**) with evaporation and crystallization only (**b**) a combined ED process for LML recovery process and (**c**) dried LML product.

**Figure 2 membranes-13-00029-f002:**
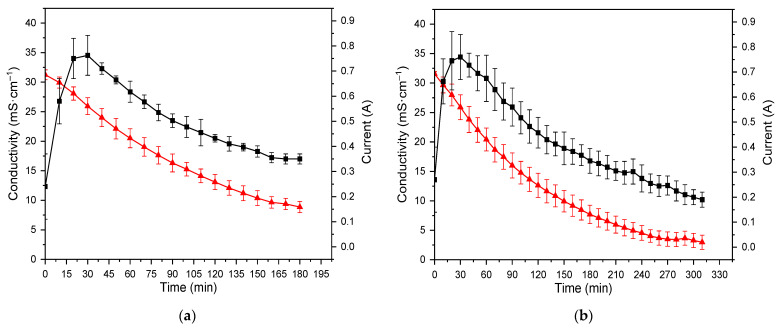
Changes in the conductivity (▲) and current (■) for (**a**) LML R70; (**b**) during ED desalination of LML R90.

**Figure 3 membranes-13-00029-f003:**
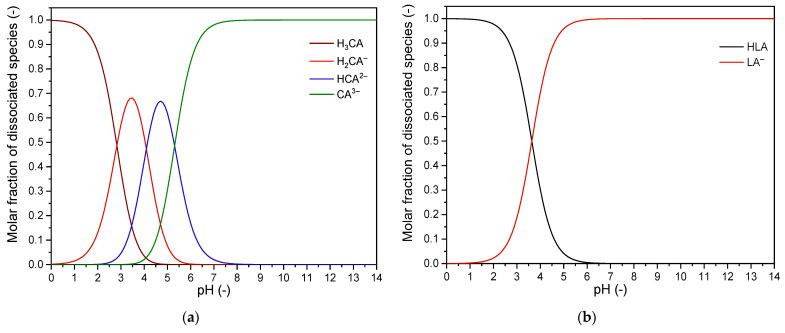
The molar fraction of (**a**) dissociated citric acid species; (**b**) dissociated lactic acid species.

**Figure 4 membranes-13-00029-f004:**
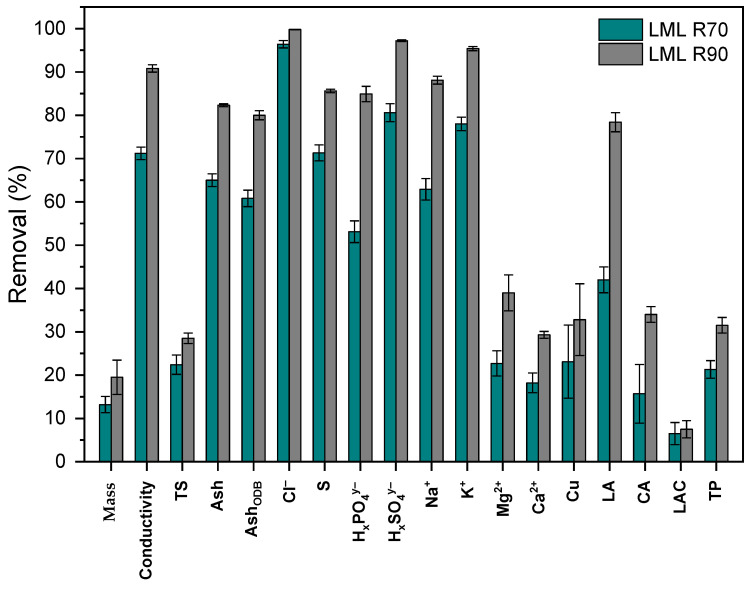
The removal of the absolute content of the components of final products LML R70 and LML R90.

**Figure 5 membranes-13-00029-f005:**
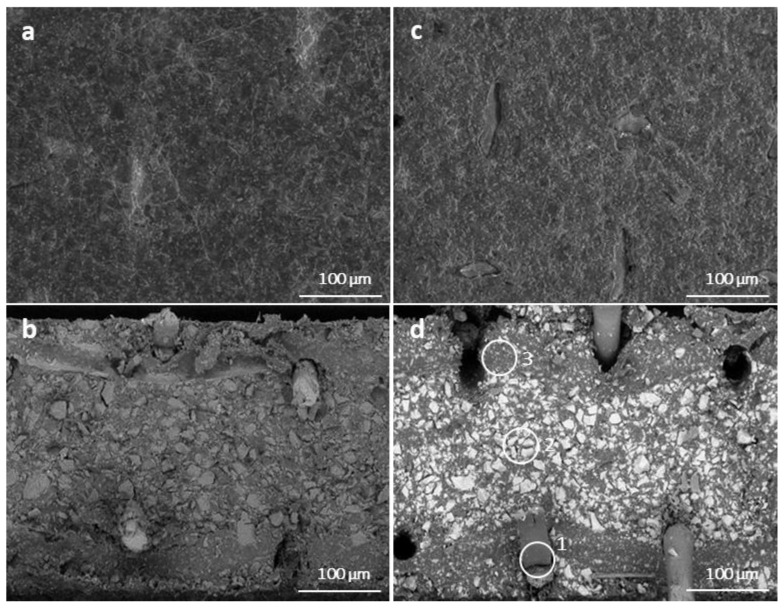
SEM micrographs images: AMH ((**a**): surface, (**b**): cross-section); CMH ((**c**): surface, (**d**): cross-section). Membrane phases: reinforcing fabric (**1**); ion exchange resin (**2**); binder polymer (**3**).

**Table 1 membranes-13-00029-t001:** Key characteristics of the flows in the ED process.

Parameter	Feed/Product	Concentrate (Brine)	Electrolyte
Initial solution	LML	Deionized water	Sodium nitrate
Initial mass, kg	1.00 ± 0.02	0.50 ± 0.02	0.25 ± 0.02
Electrical conductivity, mS·cm^−1^	31.85 ± 0.15	–	11.91 ± 1.10
Concentration, w/w%	24.58 ± 0.30	–	1.0 ± 0.1
Solution flow rates, L·h^−1^	55 ± 2	55 ± 2	50 ± 2
Linear flow velocity, cm·s^−1^	4.8 ± 0.3	4.8 ± 0.3	17.4 ± 1.0
Spacers thickness, mm	0.800± 0.005	0.800 ± 0.005	1.000 ± 0.005
Feed pH, –	5.12 ± 0.04	max. 5.5 ± 0.5	3.5 ± 1.0
Final conductivity (R70), mS·cm^−1^	9.15 ± 0.48	15.0 ± 1.5	–
Final pH (R70), –	5.07 ± 0.08	–	–
Final conductivity (R90), mS·cm^−1^	2.95 ± 0.28	15.0 ± 1.5	–
Final pH (R90), –	4.76 ± 0.16	–	–
Temperature, °C	15 ± 2	15 ± 2	15 ± 2

Data are the average of 4 repetitions with a confidence level of *p* = 0.95.

**Table 2 membranes-13-00029-t002:** Electrodialysis performance coefficients.

Parameter	Unit	Product R70	Product R90
*Κ* (conductivity)	mS·cm^−1^	9.15 ± 0.48	2.95 ± 0.28
*J* (salt transport)	g_Ash_∙m^−2^∙h^−1^	184.42 ± 13.86	144.38 ± 17.14
*E* (energy consumption)	Wh∙kg_F_^−1^	20.60 ± 1.17	30.03 ± 3.41
*C*_F_ (capacity)	kg_F_∙m^−2^∙h^−1^	5.61 ± 0.50	3.37 ± 0.40
*m*_W_ (water consumption)	kg_W_∙kg_F_^−1^	2.62 ± 0.09	3.48 ± 0.30
*m*_HNO3 65%_ (65% acid consumption)	g_Acid_∙kg_F_^−1^	1.44 ± 1.10	2.08 ± 1.59

**Table 3 membranes-13-00029-t003:** Conductivity, pH, mineral and organic composition of initial and desalted LML samples.

Parameter		Initial LML	Desalted LML (R70)	Desalted LML (R90)
Mass	kg	1.001 ± 0.001	0.868 ± 0.018	0.806 ± 0.040
Conductivity	mS·cm^−1^	31.85 ± 0.15	9.15 ± 0.48	2.95 ± 0.28
Density	g·cm^−3^	1.1235 ± 0.0007	1.1010 ± 0.0052	1.0952 ± 0.0025
pH	-	5.12 ± 0.04	5.07 ± 0.08	4.76 ± 0.16
Total solids	w/w%	24.58 ± 0.30	22.00 ± 0.72	21.83 ± 0.87
Ash	w/w%	5.147 ± 0.049	2.08 ± 0.062	1.134 ± 0.047
Ash, ODB	w/w%	20.943 ± 0.057	9.458 ± 0.257	5.196 ± 0.100
S	mg·kg^−1^	652 ± 10	216 ± 12	117 ± 7
Cu	mg·kg^−1^	0.634 ± 0.028	0.562 ± 0.060	0.529 ± 0.057
Na^+^	mg·kg^−1^	4504 ± 86	1926 ± 90	665 ± 21
K^+^	mg·kg^−1^	15,264 ± 206	3867 ± 209	872 ± 102
Mg^2+^	mg·kg^−1^	755 ± 10	673 ± 22	572 ± 59
Ca^2+^	mg·kg^−1^	3391 ± 14	3197 ± 82	2979 ± 119
Cl^−^	mg·kg^−1^	8228 ± 112	343 ± 77	19 ± 5
H_x_PO_4_^y−^	mg·kg^−1^	10,146 ± 155	5493 ± 262	1897 ± 249
H_x_SO_4_^y−^	mg·kg^−1^	1375 ± 67	307 ± 29	48 ± 4
LA	mg·kg^−1^	17,517 ± 643	11,726 ± 463	4695 ± 538
CA	mg·kg^−1^	10,203 ± 194	9914 ± 976	8360 ± 521
Lactose	g·kg^−1^	153.5 ± 2.2	165.5 ± 5.6	176.4 ± 7.8
Total protein	g·kg^−1^	18.66 ± 0.03	16.94 ± 0.47	15.87 ± 0.7

Data are the average of 4 repetitions with a confidence level of *p* = 0.95.

**Table 4 membranes-13-00029-t004:** Resistivity (R_S_) and apparent permselectivity (P) of ion-exchange membranes (IEMs).

Membranes		R_S_ (Ω·cm)	P (%)
AMH	Initial	133 ± 4	92.8 ± 0.2
	Used	116 ± 4	92.1 ± 0.2
CMH	Initial	122 ± 4	92.0 ± 0.2
	Used	124 ± 4	94.0 ± 0.2

## Data Availability

The data presented in this study are available on request from the corresponding author.

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
