# Peer review of "Lactose Mother Liquor Stream Valorisation Using an Effective Electrodialytic Process"

_membranes, 2022, doi:10.3390/membranes13010029_

Round 1

Reviewer 1 Report

This work focuses on optimizing the downstream processes before the crystallization of lactose. The process line includes a pretreatment and desalination by ED of the industrial waste solution of the lactose mother liquor (LML). Two different levels of demineralization (70% and 90%) of the LML solutions were applied to decrease the mineral components and organic acids of the LML samples. The ED performance parameters such as ash transfer rate (J), the specific capacity (CF) of the ED, and specific electric energy consumption (E) were determined, and the influence of the LML solution on the monopolar ion-exchange membranes has been investigated. I appreciate the authors' effort in compiling the data and their review. However, I request a further improvement of the authors' content to make it a more stunning and incredible piece for readers. My comments are;  

Comments

  1. Please adjust the arrangement in the introduction. Such as membrane applications should be discussed first, and their application then moves toward the main objectives of this work.
  2. Many sentences, such as "The ED is used to purify many kinds of feed, including different types of whey, wine, sugars or glycols," are either repeated or mismanaged. 
  3.  The author stated that "the removal efficiency of monovalent ions such as K+, Na+, and Cl- is higher than multivalent ions such as Ca2+, Mg2+, SO42− and PO43−.." The authors used a traditional Ion exchange membrane; how the perm-selectivity between monovalent can be obtained?
  4. The authors mentioned that "After the ED process, a decrease in resistance and permselectivity of AMHs is observed (Table 4), which can be attributed to the expansion of pores and ionic channels under the water flow pressure". After the ED process, the resistance of the membranes usually increases due to scaling how this process is different. Please comment.
  5. The English writing, spaces, abbreviations, connectivity, and format should be double-checked.

Author Response

The authors thank the reviewer for the comments that help us to clarify some questions and improve the manuscript. Please find our response to the reviewer's comments and changes in the manuscript marked in green colour.

Point 1: Please adjust the arrangement in the introduction. Such as membrane applications should be discussed first, and their application then moves toward the main objectives of this work.

Response: Thank you very much for the suggestion. This article is more related to industrial waste stream products in the food industry. That is the reason why the first part is focused on by-products and lactose production. Then, included membrane filtration and electromembrane methods. 

Point 2: Many sentences, such as "The ED is used to purify many kinds of feed, including different types of whey, wine, sugars or glycols," are either repeated or mismanaged.

Response: Changes have been implemented.

Point 3: The author stated that "the removal efficiency of monovalent ions such as K+, Na+, and Cl- is higher than multivalent ions such as Ca2+, Mg2+, SO42− and PO43− .." The authors used a traditional Ion exchange membrane; how the perm-selectivity between monovalent can be obtained?

Response: The difference in the rate of removal of individual ions is due to their difference in size and effective charge in solution. Thus, the radius of a hydrated ion is the greater, the smaller its crystallographic radius. For example, in the case of K+ and Na+, the smaller size of the sodium ion results in its higher electron density, so its hydration shell is larger than that of the potassium ion, and the effective charge is lower, which ensures more efficient removal of potassium. We have updated the article according to the comment.

Point 4: Authors mentioned that "After the ED process, a decrease in resistance and permselectivity of AMHs is observed (Table 4), which can be attributed to the expansion of pores and ionic channels under the water flow pressure". After the ED process, the resistance of the membranes usually increases due to scaling how this process is different. Please comment.

Response: After ED experiments, the LML membrane stack was cleaned sequentially (according to a supplier’s instruction) with nitric acid and sodium hydroxide solutions. Then, membranes analyses were performed. Since the resistance of the membranes did not change or even decreased, it can be concluded that even under such mild CIP cleaning conditions, possible salt deposits are removed. We have added CIP conditions to the experimental part.

Point 5: The English writing, spaces, abbreviations, connectivity, and format should be double-checked.

Response: Changes have been implemented.

Reviewer 2 Report

This study investigated the desalination of the industrial waste solution of the lactose mother liquor using ED. Even though it might be interesting for the reader, the scientific quality of the present mansucript is very poor. All the results can be expected and the conclusions are the basic knowledge of the ED community worked in this field. I cannot recommend the publication of present work. Some comments,

Table 4, the resistivity is not a common defination for an ion exchange membrane. 

Fig. 3 is redudant with very little information.

Fig. 5b and 5d, if it is the cross-section of the membranes, the membranes are too much thick based on the scale bar. 

Author Response

The authors thank the reviewer for the comments and evaluation of the article. The article is mostly focused on industrial technology and the valorization of the waste stream from lactose production. From our point of view, it is important to present the results, mostly the mineral profile of LML solutions. Based on this most dairy companies integrate the membrane process to improve the quality and product yield.

Point 1: Table 4, the resistivity is not a common defination for an ion exchange membrane. 

Response: In all the types of IEXs, the most important parameters are resistivity and permselectivity.

Point 2: Fig. 3 is redudant with very little information.

Response: We don’t think that this information is not important in this type of feed solution.

Point 3: Fig. 5b and 5d, if it is the cross-section of the membranes, the membranes are too much thick based on the scale bar.

Response: Thank you for the comment, you are right. It was changed according to the SEM scale bar.

Reviewer 3 Report

Manuscript ID: membranes-2035243 by Arthur Merkel, Matej Vavro, Ladislav ÄŒopák, Lukáš DvoÅ™ák, Lilia Ahrné, Christian Ruchti. reports on “Lactose Mother Liquor Stream Valorisation Using An Effective Electrodialytic Process”. They reported on the optimization of the downstream processes before the crystallization of lactose. The process line includes a pretreatment and desalination by ED of the industrial waste solution of the lactose mother liquor (LML). The LML was diluted to 25% total solids to overcome hydraulic issues with the ED desalination process. Two different levels of demineralization (70% and 90%) of the LML solutions were applied to decrease the mineral components and organic acids of the LML samples Although some results are brought into focus, some improvements need to do before it is considered publication due to the following key points:  

·                    The novelty of this work as compared with other studies should be addressed

·                    Did the authors check the molecular weight of LML? Did the authors observe any difficulty in the electro dialytic process?

·                    In electrodialysis, the decrease in conductivity was observed in the initial time. Could you kindly clarify.

·                    In SEM images, there is some particles distribution on the surface. the author should explain the surface and cross section morphology more detailed.

·                    Why inorganic ions decreased from 65 and 83% for LML R70 and LML R90 as compared with organic ions?

·                    For understanding the chemical interaction and composition, better to include the FTIR characterization of samples.

·                    Could you specify the pore diameter of the sample in the section 3.3

·                    Comparison of present work with previous report is missing.

·                    Please give visual demonstration for the fabrication of material

·                    English should be improved throughout the article

·                    Evaluate the effect of the ED process on the mineral and organic composition of LML

Author Response

The authors thank the reviewer for the comments that help us to clarify some questions and improve the manuscript. Please find our response to the reviewer's comments and changes in the manuscript marked in yellow colour.

Point 1: The novelty of this work as compared with other studies should be addressed.

Response: Unfortunately, there are no papers dedicated to the desalination of lactose mother liquor or using other technologies to remove inorganic and organic components. Furthermore, nowadays, commercial companies waste molasses solutions, also from beet (beet-molasses; beet sugar syrup) production as wastewater streams. In this case, it is simply impossible to compare the efficiency of processes, membrane stability, clogging etc. Conclusions present the removal efficiency of inorganic and organic components as the main factor to recover LML the re-processing.

Point 2: Did the authors check the molecular weight of LML? Did the authors observe any difficulty in the electrodialytic process?

Response: We did not observe the molecular weight of the LML solution. Mostly it depends on the pre-treatment of the solution (filtration process before the ED). On the industrial side is very difficult to have solutions with the same physical-chemical composition/properties.

Point 3: In electrodialysis, the decrease in conductivity was observed in the initial time. Could you kindly clarify.

Response: The spontaneous conductivity decrease at the initial time, when no voltage is applied on the electrodes, is attributed to ions transfer by a diffusion mechanism. Thus, ions are transported against the concentration gradient, from the diluate circuit to the concentrate circuit even if there is no presence of the electric field.

Point 4: In SEM images, there is some particles distribution on the surface. the author should explain the surface and cross section morphology more detailed.

Response: On the membrane surface no visual deposits or particles. You can see just fitting (fabrication) material. It is related to bad surface lamination. We have added information about membrane parts in Figure 5d.

Here is an example of the visible material on the membrane surface of the CMH (Ralex) membrane (image in the pdf file).

Point 5: Why inorganic ions decreased from 65 and 83% for LML R70 and LML R90 as compared with organic ions?

Response: The removal of the absolute inorganic ions content is higher than that of organic ions due to the difference in diffusion coefficients and hydrodynamic radius. Inorganic ions have higher diffusion coefficient values than organic ions, resulting in the higher removal of inorganic ions than organic ions. Moreover, we have recalculated the final values of ash content to the exact numbers (without rounding measurement results).

Point 6: For understanding the chemical interaction and composition, better to include the FTIR characterization of samples.

Response: Thank you for this point. We fully agree that FTIR analysis presents more information. At the time of our experiments on the industrial side, we did not have the possibility to obtain the FTIR method.

Point 7: Could you specify the pore diameter of the sample in the section 3.3.

Response: In heterogeneous membranes, there are two types of pores: pores formed at the boundaries, phase changes, and nanometer-sized pores formed by the widest membrane sizes in the ion exchange resin. As for the study of nanosized pores, and their size by the most common methods, such as, for example, SEM, it was found that they accumulate only in a hydrated state, and the conditions for obtaining SEM images involve shooting in a vacuum, which does not allow detecting the pore size.

Point 8: Comparison of present work with previous report is missing.

Response: Changes have been implemented. We have added information regarding the sequence of main components. However, it is many factors in the quality of feed material at least mineral composition and technological strategy of lactose production. Mostly, companies do not share process outlines.

Point 9: Please give visual demonstration for the fabrication of material.

Response: The visual demonstration is presented in Figure 5d.

Point 10: English should be improved throughout the article.

Response: Changes have been implemented.

Point 11: Evaluate the effect of the ED process on the mineral and organic composition of LML.

Response: Changes have been implemented. We have added additional information to the Conclusions section.

Round 2

Reviewer 1 Report

The authors have addressed all of my concerns. I recommend the publication of the manuscript.

Reviewer 2 Report

I still believe that the mariginal novelty and unsatified scientific quality of the present manuscript is not justified for publication.

Reviewer 3 Report

Accepted